# Leveraging Depth and Language for Open-Vocabulary Domain-Generalized Semantic Segmentation

**Siyu Chen**[1,2 †]     **Ting Han**[2,3 † *]     **Chengzheng Fu**[4 ‡]     **Changshe Zhang**[5 ‡]     **Chaolei Wang**[3 ‡]
**Jinhe Su**[1 *]     **Guorong Cai**[1]     **Meiliu Wu**[2 *]

[1] Jimei University, [2] University of Glasgow, [3] Sun Yat-sen University,
[4] Nanjing University of Aeronautics and Astronautics, [5] Xidian University,

## Abstract

Open-Vocabulary semantic segmentation (OVSS) and domain generalization in semantic segmentation (DGSS) highlight a subtle complementarity that motivates Open-Vocabulary Domain-Generalized Semantic Segmentation (OV-DGSS). OV-DGSS aims to generate pixel-level masks for unseen categories while maintaining robustness across unseen domains, a critical capability for real-world scenarios such as autonomous driving in adverse conditions. We introduce **Vireo**, **a novel single-stage framework for OV-DGSS that unifies the strengths of OVSS and DGSS for the first time.** Vireo builds upon the frozen Visual Foundation Models (VFMs) and incorporates scene geometry via Depth VFMs to extract domain-invariant structural features. To bridge the gap between visual and textual modalities under domain shift, we propose three key components: (1) GeoText Query, which align geometric features with language cues and progressively refine VFM encoder representations; (2) Coarse Mask Prior Embedding (CMPE) for enhancing gradient flow for faster convergence and stronger textual influence; and (3) the Domain-Open-Vocabulary Vector Embedding Head (DOV-VEH), which fuses refined structural and semantic features for robust prediction. Comprehensive evaluation on these components demonstrates the effectiveness of our designs. Our proposed Vireo achieves the **state-of-the-art performance and surpasses existing methods by a large margin** in both domain generalization and open-vocabulary recognition, offering a unified and scalable solution for robust visual understanding in diverse and dynamic environments. Code is available at https://github.com/SY-Ch/Vireo.

## 1 Introduction

Open-Vocabulary Domain-Generalized Semantic Segmentation (OV-DGSS) denotes the joint execution of open-vocabulary semantic segmentation (OVSS)[1, 2, 3, 4] and domain generalization in semantic segmentation (DGSS)[5, 6, 7, 8] tasks. It involves training a model that, without access to target-domain samples or annotations for novel categories, can generate pixel-wise segmentation for unseen classes while sustaining high performance across previously unseen domains (such as different cities, lighting environments, or climatic conditions). For instance, when an autonomous vehicle receives a query like "Can I park next to that bollard?", its perception system must comprehend the linguistic input and accurately segment the referenced object at the pixel level—even under adverse conditions such as dim lighting, rain-streaked lenses, or region-specific visual appearances.

OVSS and DGSS share notable similarities and can both be implemented using either multi-stage or single-stage strategies. In multi-stage OVSS [1], candidate regions or coarse masks are generated and then classified by a text approach, while multi-stage DGSS [9]first aligns domains through adversarial alignment or style transfer and then trains the segmentation approach on the aligned features.

39th Conference on Neural Information Processing Systems (NeurIPS 2025).

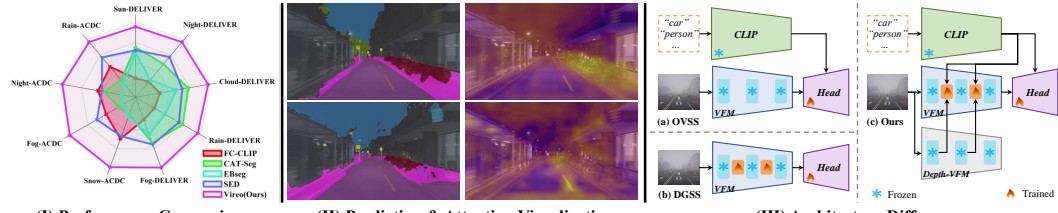

| (I) Performance Comparison | (II) Prediction & Attention Visualization | (III) Architecture Differences |

Figure 1: **Overview of Vireo and effectiveness.** (I) Performance comparison under various adverse conditions across ACDC and DELIVER datasets, showing that Vireo consistently outperforms existing methods. (II) Qualitative visualization of segmentation predictions (left) and attention maps (right) under extreme night scenes, illustrating Vireo's robustness and precise alignment with semantic cues. (III) Architectural comparison: (a) Traditional OVSS and (b) DGSS pipelines freeze or fine-tune VFM separately without cross-modal integration; (c) Our proposed Vireo introduces GeoText Query and Depth-VFM integration to enhance both semantic alignment and domain robustness.

In single-stage OVSS [10, 11], the segmentation head is dynamically conditioned on text prompts to directly produce masks for each class. In contrast, single-stage DGSS [12, 13] incorporates domain-invariant modules within the backbone or segmentation head, enabling the model to jointly learn segmentation and generalization in a unified forward pass without requiring distinct stages. The key distinction lies in their focus: OVSS requires integration of visual features with textual semantics to accurately identify unseen classes, whereas DGSS emphasizes robustness to domain shifts.

Consequently, integrating open-vocabulary recognition of novel classes with domain robustness into a unified framework presents two primary challenges: (1) *Text–vision alignment modules often degrade outside the source domain, leading to significant performance drops even for previously seen classes.* (2) *Domain-invariant strategies may suppress fine-grained semantic cues, hindering the model's ability to precisely respond to detailed textual queries.*

Recent single-stage DGSS studies have increasingly adopted strategies that fine-tune learnable tokens across the layers of a frozen Visual Foundation Model (VFM) [14, 15, 16, 17, 18] to adapt its feature representations. In OVSS, the VFM encoder is typically fully frozen, with efforts focused on designing the decoder to endow the model with open-vocabulary recognition capabilities. This reveals a subtle complementarity between the two paradigms: **DGSS emphasizes the encoder by leveraging the VFM's strong feature generalization to learn cross-domain representations, whereas OVSS freezes the encoder and emphasizes the decoder to enable open-vocabulary recognition.**

Moreover, in cross-domain scenarios, depth and geometric cues are largely insensitive to variations in illumination and texture[19, 20]. They supply reliable spatial constraints, easing the distribution shift of RGB features and sharpening boundary localization. Recent studies such as DepthForge [21] have shown that injecting depth query into a frozen VFM boosts domain generalization. Motivated by these findings, we adopt DepthAnything V2 [22] as our Depth VFM: its diverse pre-training delivers consistent depth estimation across domains, and keeping it fully frozen incurs minimal training cost while preserving real-time inference speed.

In this paper, we propose a VFM-based single-stage pipeline for OV-DGSS, termed **Vireo**. Specifically, in the encoder, both the VFM and DepthAnything modules are kept frozen. The VFM is leveraged to robustly capture cross-domain visual features, while DepthAnything extracts the scene's intrinsic geometric structure. On this basis, we introduce **Geometric-Text Query (GeoText Query)** to fuse the extracted structural features with manually provided textual cues, progressively refining the feature maps between the frozen VFM layers. To mitigate the slow convergence caused by sparse gradients propagating through the frozen encoder, we introduce a **Coarse Mask Prior Embedding (CMPE)** at the beginning of the decoder to inject denser gradient signals. This design accelerates the convergence of mask supervision and further strengthens the influence of textual priors. Subsequently, we design the **Domain-Open-Vocabulary Vector Embedding Head (DOV-VEH)** to strengthen the synergistic integration of structural and textual modalities, ensuring that both cross-domain structural features and open-vocabulary cues learned by the GeoText Query are fully utilized in the final prediction.

**For challenge 1**: We find that GeoText Query not only capture structural and semantic cues within the frozen VFM encoder but also guide the progressive refinement of its feature representations. **For challenge 2**: The CMPE enhances gradient back-propagation into the encoder, while the

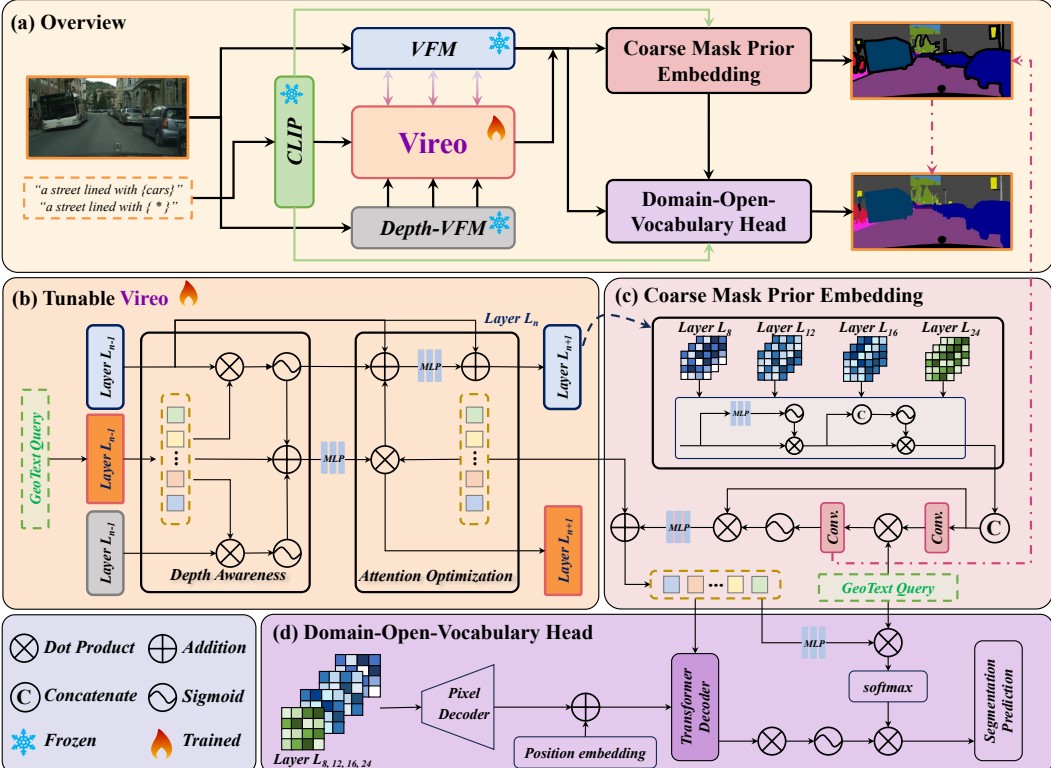

Figure 2: **Overview of the proposed Vireo framework for OV-DGSS.** (a) **Framework Overview.** Geometric-Text Query permeate our model: they're injected in Tunable Vireo to align domain priors, reused in CMPE to guide multi-scale feature fusion, and employed in DOV-VEH as queries for open-vocabulary segmentation, forming a unified end-to-end loop. (b) **Tunable Vireo.** GeoText Query are used to inject structural-textual priors across multiple layers. Depth-aware fusion and attention optimization are applied between intermediate frozen layers to progressively refine representations. (c) **Coarse Mask Prior Embedding (CMPE).** Multi-scale features from VFM are combined with GeoText Query to provide dense supervision and gradient signals for downstream modules. (d) **Domain-Open-Vocabulary Head (DOV-VEH).** Multi-level features are processed by a Pixel Decoder and Transformer Decoder to produce final predictions guided by open-vocabulary text queries.

redesigned DOV-VEH deepens the fusion of visual and textual priors. These components form a unified framework that simultaneously achieves domain robustness and strong open-vocabulary generalization. Our main contributions are summarized as follows:

(1) We propose **Vireo**, a novel VFM-based single-stage framework for OV-DGSS.

(2) We introduce **GeoText Query** to progressively refine frozen VFM features by injecting geometric cues from DepthAnything and aligning them with textual semantics, enabling structural-semantic fusion across encoder layers.

(3) We design two complementary modules—**CMPE** for enhancing gradient flow and **DOV-VEH** for fusing visual and textual priors—together boosting segmentation performance under domain shifts and unseen classes.

## 2 Related Works

**Open-Vocabulary Semantic Segmentation**. Open-Vocabulary Semantic Segmentation (OVSS) aims to segment objects based on arbitrary textual descriptions, moving beyond fixed, pre-defined categories. A key enabler for OVSS [23, 24, 1, 25] has been vision-language models (VLMs), particularly CLIP [16], which aligns visual and textual representations.

**Domain-Generalized Semantic Segmentation**. Domain Generalization in Semantic Segmentation (DGSS) addresses the performance degradation when models encounter unseen target domains due to domain shift—variations in data distributions (e.g., lighting, weather). Data augmentation [26, 27] and learning domain-invariant representations [28, 29] are two key strategies to enhance model robustness. This detailed discussion in the appendix further contextualizes our specific contributions within the broader landscape of these research areas.

## 3   Methodology - Vireo

We formulate the open-vocabulary DGSS problem as predicting a pixel-wise segmentation mask from an input image and a set of text labels. Given an image $I$ and a set of open-vocabulary class names $C$, our goal is to produce a mask $M$ where each pixel is classified according to one of the classes in $C$. As illustrated in Figure 2, our framework Vireo tackles this by leveraging frozen vision and text encoders with additional trainable prompt and depth modules for improved generalization, it consists of three key modules tailored for the OV-DGSS: (1) **Tunable Vireo with GeoText Query:** Introduces GeoText Query to inject and refine both geometric and textual information across layers of a frozen VFM. (2) **Coarse Mask Prior Embedding (CMPE):** Generates coarse prior masks to guide segmentation and reinforce gradient flow from the decoder back to the frozen encoder. (3) **Domain-Open-Vocabulary Vector Embedding Head (DOV-VEH):** Integrates visual, geometric, and semantic features to produce final OV-DGSS predictions.

### 3.1   Overview

The input RGB image $I \in \mathbb{R}^{H \times W \times 3}$ is duplicated and independently fed into two frozen encoders: a VFM encoder $\mathcal{F}^V$ and a DepthAnything encoder $\mathcal{F}^D$. These backbones extract a series of multi-scale features $f_l^V$ and $f_l^D$ respectively, where $l$ denotes the encoder layer index in $[1, L]$. In parallel, a set of class labels $\mathcal{C} = \{c_1, \ldots, c_K\}$ is transformed into natural language prompts $T = \{\text{prompt}_1, \ldots, \text{prompt}_K\}$, which are encoded using a frozen CLIP text encoder $\mathcal{F}^T$ to obtain textual embeddings $t_k \in \mathbb{R}^d$. These features serve as shared semantic priors across the entire framework and are precomputed once at initialization. In the Tunable Vireo module, each layer receives a tuple $(f_l^V, f_l^D, t_k)$, and applies the GeoText Query $P_l$ to align and refine the visual features. This is achieved by computing cross-modal attention maps $\mathcal{A}_l = \text{Attn}(P_l, f_l^V, f_l^D, t_k)$ followed by feature fusion and projection. The refined output $\hat{f}^V{}_l$ is then forwarded back to update the VFM's layer-wise activations.

We select the refined visual features $\{f_{l_i}^V\}_{i=1}^4$ from the VFM encoder at layers $l_1 = 8$, $l_2 = 12$, $l_3 = 16$, and $l_4 = 24$ for downstream decoding. In the Coarse Mask Prior Embedding (CMPE) module, these features are first upsampled to a unified spatial resolution and passed through a channel-spatial attention gating function $\mathcal{G}(\cdot)$. The gated outputs are then fused to form a global coarse feature $f^M \in \mathbb{R}^d$. The $f^M$ is projected and matched with the text embeddings $t_k \in \mathbb{R}^d$ to produce a coarse class probability map $\mathcal{M} \in \mathbb{R}^{H \times W \times K}$, where $\mathcal{M}$ serves both as a weak supervision signal and as a prior to construct the query embeddings $\mathcal{Q}$. The query prior is added to the GeoText Query and forwarded to the final segmentation head. In the Domain-Open-Vocabulary Vector Embedding Head (DOV-VEH), the multi-scale features $\{\hat{f}_{l_i}^V\}_{i=1}^4$ are passed through a pixel decoder $\mathcal{D}_p(\cdot)$ to enhance spatial representation, followed by a Transformer decoder $\mathcal{D}_T(\cdot)$ that leverages positional embedding. The GeoText Query serve as learnable queries and interact with both the decoded features and the text embeddings, producing pixel-level mask embeddings $\mathcal{E}_{\text{mask}}(x, y) \in \mathbb{R}^d$ and classification embeddings $\mathcal{E}_{\text{cls}}(k) \in \mathbb{R}^d$. The final prediction is $\hat{\mathcal{M}}(x, y, k) \in \mathbb{R}^{H \times W \times K}$ provides the pixel-wise semantic prediction with both fine-grained detail and open-vocabulary generalization capability.

### 3.2   Tunable Vireo with GeoText Query

To improve efficiency, we first precompute and share the textual prompt embeddings using a frozen CLIP text encoder $\mathcal{F}^T$. Specifically, a set of class labels $\mathcal{C} = \{c_1, \ldots, c_K\}$ is transformed into language prompts $T = \{\text{prompt}_1, \ldots, \text{prompt}_K\}$, which are encoded as $t_k = \mathcal{F}^T(\text{prompt}_k)$, where $t_k \in \mathbb{R}^d$. These embeddings are reused across all GeoText Prompt layers, CMPE, and DOV-VEH modules to avoid redundant computation.

During inference, the input image $I \in \mathbb{R}^{H \times W \times 3}$ is simultaneously processed by a frozen visual encoder $\mathcal{F}^V$ and a frozen depth encoder $\mathcal{F}^D$ (e.g., DepthAnything). For each selected layer $l \in \{1, \dots, L\}$, we obtain the visual feature map $f_l^V = \mathcal{F}_l^V(I)$ and the depth feature map $f_l^D = \mathcal{F}_l^D(I)$. Each layer of the Tunable Vireo module receives the tuple $(f_l^V, f_l^D, \{t_k\})$ along with a layer-specific GeoText Prompt $P_l \in \mathbb{R}^{N \times d}$, where $N$ is the number of learnable queries.

The prompt $P_l$ first interacts with the textual embeddings via a fusion block, then attends to both $f_l^V$ and $f_l^D$ through cross-attention mechanisms:

$$\mathcal{A}_l = \text{CrossAttn}(P_l, \ f_l^V, \ f_l^D, \ \{t_k\}). \tag{1}$$

The attention outputs are fused via weighted summation, then passed through an MLP projection layer and multiplied element-wise with $P_l$. A residual connection adds this result to the original feature map $f_l^V$, yielding a refined visual representation $\hat{f}_l^V$. Finally, another MLP transforms $\hat{f}_l^V$ into the input for the next VFM layer, and the updated GeoText Prompt $P_{l+1}$ is passed forward.

This progressive refinement continues across all selected layers, enabling the model to inject and align geometric and semantic information at multiple scales, thereby enhancing cross-domain robustness and open-vocabulary generalization.

### 3.3  Coarse Mask Prior Embedding (CMPE)

We select the refined visual features $\{\hat{f}_{l_i}^V\}_{i=1}^4$ from the VFM encoder at layers $l_1 = 8$, $l_2 = 12$, $l_3 = 16$, and $l_4 = 24$, respectively. Each feature map is upsampled to a common spatial resolution $(H \times W)$ via bilinear interpolation, and then passed through an Adaptive Attention Gate (AAG) $\mathcal{G}(\cdot)$, which enhances informative channels and spatial regions. Specifically, AAG applies two $1 \times 1$ convolutions followed by ReLU and Sigmoid for channel attention, and a $3 \times 3$ convolution followed by Sigmoid for spatial attention.

The attended features are concatenated along the channel axis and fused via a $1 \times 1$ convolution to restore the embedding dimension $d$, yielding a fused feature representation: $f^M = \text{Fuse}(\mathcal{G}(\hat{f}_{l_i}^V))$. We apply a residual addition between $f^M$ and the final layer output $\hat{f}_{l_4}^V$ to obtain the updated mask feature: $f^M = f^M + \hat{f}_{l_4}^V$. This fused feature $f^M(x, y) \in \mathbb{R}^d$ is projected to the same dimension as the text embeddings $t_k \in \mathbb{R}^d$ and compared via Einstein summation to generate a coarse semantic probability map $\mathcal{M}(x, y, k) = \langle f^M(x, y), t_k \rangle$, where $\mathcal{M} \in \mathbb{R}^{B \times K \times H \times W}$. This coarse mask is supervised with a segmentation loss to enhance gradient flow through the frozen encoder.

To generate query priors for the downstream segmentation head, we first normalize $\mathcal{M}$ across the spatial domain to derive attention weights:

$$\alpha_k(x, y) = \frac{\exp(\mathcal{M}(x, y, k))}{\sum_{x', y'} \exp(\mathcal{M}(x', y', k))}. \tag{2}$$

Then, we compute the class-specific aggregated feature by spatially weighting $f^M$:

$$f_k^{\text{class}} = \sum_{x,y} \alpha_k(x, y) \cdot f^M(x, y). \tag{3}$$

Each $f_k^{\text{class}}$ is projected into the embedding space as $e_k^{\text{class}} \in \mathbb{R}^d$, and combined with a set of learnable queries vectors $\{q_j\}_{j=1}^{N_q}$ to produce the query priors:

$$q_j^{\text{prior}} = \sum_{k=1}^K \text{Softmax}(\langle q_j, \ e_k^{\text{class}} \rangle) \cdot e_k^{\text{class}}. \tag{4}$$

The final query priors $\{q_j^{\text{prior}}\}_{j=1}^{N_q}$ are added to the corresponding GeoText Query and forwarded into the Domain-Open-Vocabulary Vector Embedding Head (DOV-VEH).

### 3.4  Domain-Open-Vocabulary Vector Embedding Head (DOV-VEH)

The DOV-VEH module receives the refined multi-scale features $\{\hat{f}_{l_i}^V\}_{i=1}^4$ from the VFM encoder, where $l_1 = 8$, $l_2 = 12$, $l_3 = 16$, and $l_4 = 24$, along with the updated GeoText Query $\{P_l\}$. These

Table 1: Performance comparison between our Vireo and existing OVSS and DGSS methods under *Citys.* → *ACDC + BDD. + Map.*, and *GTA5.* → *Citys. + BDD. + Map.* generalization settings. Top three results are highlighted as best , second , and third , respectively. (%)

| Method | Proc. & Year | Trained on Cityscapes | | | | | | | Trained on GTA5 | | |
| --- | --- | --- | --- | --- | --- | --- | --- | --- | --- | --- | --- |
| | | Night-ACDC | Fog-ACDC | Rain-ACDC | Snow-ACDC | BDD 100k | Mapillary | GTA5 | Cityscapes | BDD100k | Mapillary |
| **OVSS Method** | | | | | | | | | | | |
| FC-CLIP [30] | NeurIPS2023 | 40.8 | 64.4 | 63.2 | 61.5 | 55.92 | 66.12 | 47.12 | 53.54 | 51.41 | 58.60 |
| EBSeg [31] | CVPR2024 | 27.7 | 56.5 | 51.8 | 50.1 | 48.91 | 63.40 | 42.61 | 44.80 | 40.59 | 56.28 |
| CAT-Seg [10] | CVPR2024 | 37.2 | 58.3 | 45.6 | 49.0 | 48.26 | 54.74 | 45.18 | 43.52 | 44.28 | 50.88 |
| SED [3] | CVPR2024 | 38.7 | 69.0 | 56.4 | 60.2 | 53.30 | 64.32 | 48.93 | 47.45 | 48.16 | 57.38 |
| **DGSS Method** | | | | | | | | | | | |
| *ResNet based:* | | | | | | | | | | | |
| IBN [32] | ECCV2018 | 21.2 | 63.8 | 50.4 | 49.6 | 48.56 | 57.04 | 45.06 | - | - | - |
| RobustNet [9] | CVPR2021 | 24.3 | 64.3 | 56.0 | 49.8 | 50.73 | 58.64 | 45.00 | 36.58 | 35.20 | 40.33 |
| WildNet [33] | CVPR2022 | 12.7 | 41.2 | 34.2 | 28.4 | 50.94 | 58.79 | 47.01 | 44.62 | 38.42 | 46.09 |
| *Transformer based:* | | | | | | | | | | | |
| HGFormer [34] | CVPR2023 | 52.7 | 69.9 | 72.0 | 68.6 | 53.40 | 66.90 | 51.30 | - | - | - |
| CMFormer [35] | AAAI2024 | 33.7 | 77.8 | 67.6 | 64.3 | 59.27 | 71.10 | 58.11 | 55.31 | 49.91 | 60.09 |
| *VFM based:* | | | | | | | | | | | |
| REIN [36] | CVPR2024 | 55.9 | 79.5 | 72.5 | 70.6 | 63.54 | 74.03 | 62.41 | 66.40 | 60.40 | 66.10 |
| FADA [37] | NeurIPS2024 | 57.4 | 80.2 | 75.0 | 73.5 | 65.12 | 75.86 | 63.78 | 68.23 | 61.94 | 68.09 |
| **OV-DGSS Method** | | | | | | | | | | | |
| **Vireo (Ours)** | - | 60.6 | 82.3 | 76.3 | 76.2 | 66.73 | 75.99 | 67.86 | 70.69 | 62.91 | 69.63 |

features are first processed by a pixel decoder $\mathcal{D}_p(\cdot)$, which leverages multi-scale cross-attention to extract rich spatial context: $f^{\text{pix}} = \mathcal{D}_p\left(\hat{f}_{l_i}^V\right)$. The fused feature $f^{\text{pix}} \in \mathbb{R}^{H \times W \times d}$ is then compressed via a $1 \times 1$ convolution and enriched with sinusoidal positional encoding to preserve spatial structure.

The enhanced features are fed into a Transformer Decoder $\mathcal{D}_T(\cdot)$, where the GeoText Query act as learnable queries. Through stacked layers of self-attention and cross-attention with $f^{\text{pix}}$, the model captures fine-grained visual-semantic alignment, yielding a set of high-resolution mask features $\mathcal{E}_{\text{mask}}(x, y) \in \mathbb{R}^d$ at each spatial position.

Simultaneously, the GeoText Query $\{P\}$ are passed through a two-layer MLP and then interact with the text embeddings $\{t_k\}$ (precomputed from the CLIP text encoder) to produce classification-level representations $\mathcal{E}_{\text{cls}}(k) \in \mathbb{R}^d$. The final segmentation prediction $\hat{\mathcal{M}} \in \mathbb{R}^{H \times W \times K}$ is generated via an Einstein summation over the two embeddings:

$$\hat{\mathcal{M}}(x, y, k) = \sum_{d=1}^{D} \mathcal{E}_{\text{mask}}(x, y, d) \cdot \mathcal{E}_{\text{cls}}(k, d), \tag{5}$$

where $D$ is the feature embedding dimension. This design enables DOV-VEH to generate pixel-level segmentation masks that are both spatially accurate and semantically aligned with open-vocabulary textual queries.

## 4 Experiments

### 4.1 Datasets & Evaluation Protocols

We evaluate Vireo on six real-world datasets (**Cityscapes** [38], **BDD100K** [39], **Mapillary** [40], **ACDC** [41], **ADE150** [42], and **ADE847** [42]) and two synthetic datasets (**GTA5** [43] and **DELIVER** [44]). **Cityscapes** (City.) is an autonomous-driving dataset with 2,975 training images and 500 validation images, each at a resolution of $2048 \times 1024$. **BDD100K** (BDD.) and **Mapillary** (Map.) provide 1,000 and 2,000 validation images, respectively, at resolutions of $1280 \times 720$ and $1920 \times 1080$. **ACDC** offers 406 validation images captured under extreme conditions (night, snow, fog, and rain), each at $1920 \times 1080$. **GTA5** is a synthetic dataset that contains 24,966 labeled images obtained from a video game. **DELIVER** is a multimodal synthetic dataset comprising 3,983 training images, 2,005 validation images, and 1,897 test images across five weather conditions (cloudy, foggy, night, rainy, and sunny); every image is $1042 \times 1042$ and there are 25 classes. **ADE150** and **ADE847** refer to subsets of the ADE20K dataset [42], each containing 2,000 validation images of variable resolution sourced from diverse scenes such as SUN and Places, covering 150 and 847 semantic categories, respectively.

Following the existing DGSS evaluation protocol, we train on one dataset as the source domain and validate on multiple unseen target domains. The three standard evaluation setups are: (1) Cityscapes → ACDC; (2) GTA5 → Cityscapes, BDD100K, Mapillary; (3) Cityscapes → BDD100K, Mapillary, GTA5. To assess our proposed OV-DGSS approaches and compare its open-vocabulary capability

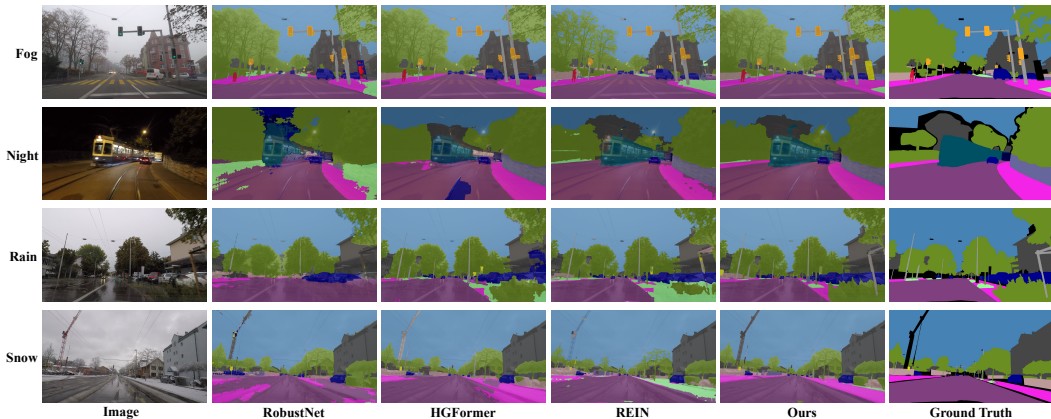

Figure 3: Key segmentation examples of existing DGSS methods and Vireo under the Citys. → ACDC unseen target domains under Night, Snow, Rain, and Fog conditions.

Table 2: Performance comparison between our Vireo and existing OVSS methods under *Citys.* → *DELIVER + ADE150 + ADE847* generalization setting. Top three results are highlighted as best , second , and third , respectively. (%)

| Method | Proc. & Year | Sun-DELIVER | Night-DELIVER | Cloud-DELIVER | Rain-DELIVER | Fog-DELIVER | ADE150 | ADE847 |
|---|---|---|---|---|---|---|---|---|
| **Train on Cityscapes** | | | | | | | | |
| *OVSS Method:* | | | | | | | | |
| FC-CLIP [30] | NeurIPS2023 | 16.93 | 14.93 | 17.50 | 16.59 | 17.26 | 16.12 | 6.29 |
| EBSeg [31] | CVPR2024 | 26.41 | 15.50 | 22.62 | 20.35 | 22.00 | 12.75 | 3.75 |
| CAT-Seg [10] | CVPR2024 | 28.21 | 20.56 | 26.22 | 26.53 | 24.80 | 20.19 | 6.95 |
| SED [3] | CVPR2024 | 27.14 | 22.79 | 24.40 | 25.18 | 25.25 | 18.86 | 5.45 |
| *OV-DGSS Method:* | | | | | | | | |
| Vireo (Ours) | – | 35.73 | 27.51 | 32.34 | 31.80 | 32.72 | 21.37 | 7.31 |
| **Train on GTA** | | | | | | | | |
| *OVSS Method:* | | | | | | | | |
| FC-CLIP [30] | NeurIPS2023 | 22.24 | 18.58 | 18.50 | 16.59 | 19.12 | 15.47 | 5.73 |
| EBSeg [31] | CVPR2024 | 32.32 | 20.05 | 26.19 | 26.19 | 28.69 | 11.87 | 4.19 |
| CAT-Seg [10] | CVPR2024 | 28.59 | 23.49 | 27.31 | 27.94 | 27.66 | 20.45 | 7.18 |
| SED [3] | CVPR2024 | 26.56 | 21.18 | 24.95 | 24.58 | 26.17 | 19.57 | 6.80 |
| *OV-DGSS Method:* | | | | | | | | |
| Vireo (Ours) | – | 38.49 | 29.89 | 33.89 | 33.46 | 35.80 | 21.23 | 7.68 |

against OVSS approaches, we additionally introduce two more configurations: (4) Cityscapes → DELIVER, ADE150, ADE847; (5) GTA5 → DELIVER, ADE150, ADE847. The evaluation metric is mean Intersection over Union (mIoU).

## 4.2 Deployment Details & Parameter Settings

Our implementation is built upon the MMSegmentation [45] codebase. We employ the AdamW optimizer with an initial learning rate of 1e-4, a weight decay of 0.05, epsilon set to 1e-8, and beta parameters of (0.9, 0.999). The total number of training iterations is 40,000, matching REIN, and we adopt a polynomial learning-rate decay schedule that reduces the learning rate to zero over 40,000 iterations with a decay power of 0.9 and no epoch-based warmup. Data augmentation comprises multi-scale resizing, random cropping (with fixed crop size and category-ratio constraint), random horizontal flipping, and photometric distortion. All experiments are conducted on an NVIDIA RTX A6000 GPU with a batch size of 8, taking approximately 14 hours to train and peaking at around 45 GB of GPU memory usage.

## 4.3 Performance Comparison

**Domain Generalization Ability**: Table 1 summarizes the evaluation results of various state-of-the-art open-vocabulary semantic segmentation (OVSS) and domain-generalized semantic segmentation (DGSS) methods under two cross-domain settings (Cityscapes → ACDC, BDD100k, Mapillary, GTA5 and GTA5 → Cityscapes, BDD100k, Mapillary). The results demonstrate that our approach achieves outstanding performance across all target datasets, significantly outperforming other OVSS/DGSS methods. Furthermore, the visualizations in Figure 3 shows that Vireo delivers

Table 3: Comparison of Seen and Unseen Category mIoU Across Weather Scenarios and Methods.

| Class | Cloud | | | | | Fog | | | | | Night | | | | | Rain | | | | | Sun | | | | |
|---|---|---|---|---|---|---|---|---|---|---|---|---|---|---|---|---|---|---|---|---|---|---|---|---|---|
| | Ours | SED | CAT-Seg | FC-CLIP | EBSeg | Ours | SED | CAT-Seg | FC-CLIP | EBSeg | Ours | SED | CAT-Seg | FC-CLIP | EBSeg | Ours | SED | CAT-Seg | FC-CLIP | EBSeg | Ours | SED | CAT-Seg | FC-CLIP | EBSeg |
| Seen | 49.14 | 36.72 | 43.92 | 28.07 | 38.10 | 49.59 | 38.28 | 41.43 | 26.06 | 35.76 | 41.63 | 34.99 | 34.89 | 23.03 | 26.03 | 47.15 | 37.35 | 41.43 | 25.59 | 31.96 | 55.16 | 41.68 | 45.10 | 26.52 | 42.63 |
| Unseen | 10.96 | 8.74 | 3.69 | 4.05 | 2.92 | 12.15 | 8.40 | 3.54 | 4.32 | 4.48 | 9.52 | 7.27 | 2.30 | 4.09 | 2.09 | 12.27 | 8.95 | 5.81 | 4.60 | 5.57 | 11.46 | 8.12 | 4.91 | 4.73 | 6.14 |
| Mean | 33.89 | 24.40 | 26.22 | 17.50 | 22.62 | 35.80 | 25.25 | 24.80 | 17.26 | 22.00 | 29.89 | 22.79 | 20.56 | 14.93 | 15.50 | 33.46 | 25.18 | 26.53 | 16.59 | 20.35 | 38.49 | 27.14 | 28.21 | 16.93 | 26.41 |

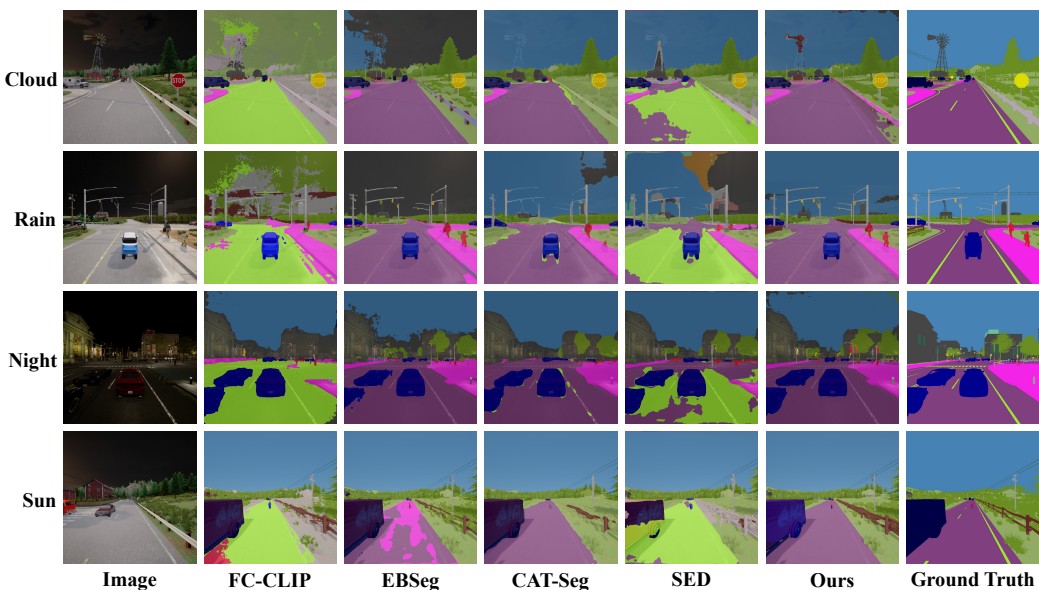

Figure 4: Key segmentation examples of existing OV-DGSS methods comparison on unseen classes and cross domain on Cityscapes → DELIVER.

satisfactory predictions in both extreme weather conditions and scenes with dense pedestrian and vehicular traffic.

**Open-Vocabulary Capability**: Table 2 presents a comparison between Vireo and other open-vocabulary semantic segmentation (OVSS) methods under the Cityscapes → DELIVER (sun, rain, night, cloud, fog), ADE150, and ADE847 configurations. The results indicate that conventional OVSS approaches suffer a sharp performance drop in extreme scenarios (e.g., night), whereas our model—enhanced by depth-based geometric features—maintains robust performance, outperforming the strongest OVSS baseline by at least 5%. Furthermore, as shown in Figure 4, the coexistence of new open classes and extreme weather conditions in the DELIVER dataset leads OVSS approaches to exhibit significant false positives and false negatives.

Table 3 compares Vireo trained on Cityscapes with other open-vocabulary semantic-segmentation methods on the DELIVER dataset across five weather conditions for both Seen and Unseen categories. Vireo achieves the highest mIoU for both groups: it outperforms the second-best method by roughly 7–10 percentage points on Seen classes and by 2–3 points on Unseen ones. Although all methods register lower mIoU on Unseen categories—underscoring the difficulty of open-vocabulary segmentation—Vireo substantially reduces this gap, confirming that its depth-geometry guidance and cross-domain alignment improve recognition of new classes. Overall, Vireo remains consistently superior across weather scenarios and category types, demonstrating stronger out-of-domain generalization.

## 4.4 Ablation Study

**Robust Performance Gains.** In Table 4, we compare parameter overhead and mIoU of lightweight tuning methods on EVA02-Large and DINOv2-Large under the GTA5→Cityscapes + BDD + Mapillary transfer. With only ≈ 3.8 M trainable parameters, Vireo leads both: 66.0% on EVA02 (+1.1% over FADA) and 67.7% on DINOv2 (+1.6%). Other schemes (REIN, LoRA, VPT, AdaptFormer)

Table 4: Performance comparison of the proposed **Vireo** against other DGSS methods under the *GTA5→Citys.+BDD.+Map.* setting.

|  | EVA02 (Large) [46, 47] | | | | |  | DINOv2 (Large) [54] | | | | |
|---|---|---|---|---|---|---|---|---|---|---|---|
| **Fine-tune Method** | Trainable Params* | **mIoU** | | | **Avg.** | **Fine-tune Method** | Trainable Params* | **mIoU** | | | **Avg.** |
|  |  | Citys. | BDD. | Map. |  |  |  | Citys. | BDD. | Map. |  |
| Full | 304.24M | 62.1 | 56.2 | 64.6 | 60.9 | Full | 304.20M | 63.7 | 57.4 | 64.2 | 61.7 |
| +AdvStyle[48] | 304.24M | 63.1 | 56.4 | 64.0 | 61.2 | +AdvStyle[48] | 304.20M | 60.8 | 58.0 | 62.5 | 60.4 |
| +PASTA[49] | 304.24M | 61.8 | 57.1 | 63.6 | 60.8 | +PASTA[49] | 304.20M | 62.5 | 57.2 | 64.7 | 61.5 |
| +GTR-LTR[50] | 304.24M | 59.8 | 57.4 | 63.2 | 60.1 | +GTR-LTR[50] | 304.20M | 62.7 | 57.4 | 64.5 | 61.6 |
| Freeze | 0.00M | 56.5 | 53.6 | 58.6 | 56.2 | Freeze | 0.00M | 63.3 | 56.1 | 63.9 | 61.1 |
| +AdvStyle[48] | 0.00M | 51.4 | 51.6 | 56.5 | 53.2 | +AdvStyle[48] | 0.00M | 61.5 | 55.1 | 63.9 | 60.1 |
| +PASTA[49] | 0.00M | 57.8 | 52.3 | 58.5 | 56.2 | +PASTA[49] | 0.00M | 62.1 | 57.2 | 64.5 | 61.3 |
| +GTR-LTR[50] | 0.00M | 52.5 | 52.8 | 57.1 | 54.1 | +GTR-LTR[50] | 0.00M | 60.2 | 57.7 | 62.2 | 60.0 |
| +LoRA[51] | 1.18M | 55.5 | 52.7 | 58.3 | 55.5 | +LoRA[51] | 0.79M | 65.2 | 58.3 | 64.6 | 62.7 |
| +AdaptFormer[52] | 3.17M | 63.7 | 59.9 | 64.2 | 62.6 | +AdaptFormer[52] | 3.17M | 64.9 | 59.0 | 64.2 | 62.7 |
| +VPT[53] | 3.69M | 62.2 | 57.7 | 62.5 | 60.8 | +VPT[53] | 3.69M | 65.2 | 59.4 | 65.5 | 63.3 |
| +REIN[36] | 2.99M | 65.3 | 60.5 | 64.9 | 63.6 | +REIN[36] | 2.99M | 66.4 | 60.4 | 66.1 | 64.3 |
| +FADA[37] | 11.65M | 66.7 | 61.9 | 66.1 | 64.9 | +FADA[37] | 11.65M | 68.2 | 62.0 | 68.1 | 66.1 |
| **+Vireo (Ours)** | 3.78M | **68.5** | **62.1** | **67.4** | **66.0** | **+Vireo (Ours)** | 3.78M | **70.7** | **62.9** | **69.6** | **67.7** |

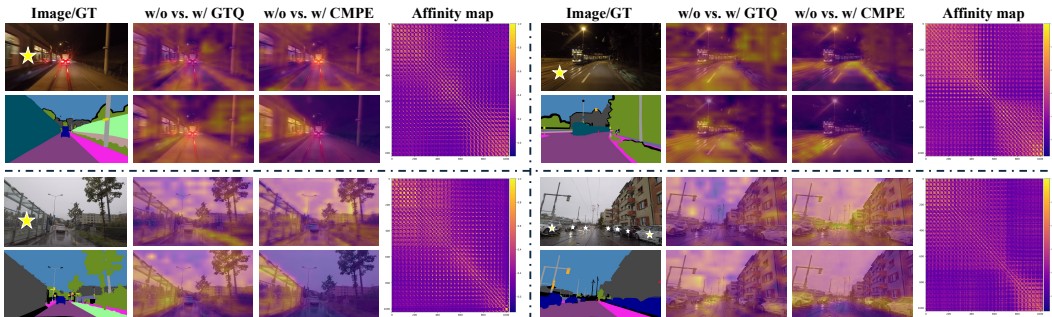

Figure 5: Visualizations of attention maps and affinity maps under differnet scenes, where CMPE denotes the Coarse Mask Prior Embedding, GTQ represents the GeoText Query, respectively.

improve a frozen backbone but fall short of Vireo's accuracy-efficiency balance. Versus full fine-tuning, Vireo slashes training cost and further improves mIoU, showing its depth-geometry queries generalize across VFMs.

Table 6 reports the parameter overhead and mIoU of Vireo and several lightweight fine-tuning schemes on four backbones (CLIP-L, SAM-H, EVA02-L, and DINOv2-L). Compared with the prompt-based REIN baseline, Vireo adds only about 0.79 M extra parameters yet achieves the highest average mIoU on every backbone: the gain is most pronounced on the parameter-constrained CLIP-L, and Vireo still surpasses heavier adapters such as FADA by 1–2 mIoU on the larger EVA02-L and DINOv2-L models, underscoring its parameter efficiency and scalability across backbones.

In Table 5, using Depth Anything V2 alone yields an approximate 0.6% mIoU gain across all six scenarios, and adding depth augmentation with attention optimization (DA + AO) delivers a further ≈ 1% improvement. The DOV-VEH and CMPE modules each add 0.5%–0.8%, validating mask vectors and dense gradient embedding. GeoText Query on its own provides a substantial ≈ 4.4% boost, underscoring the complementary benefits of fusing semantic and geometric cues.

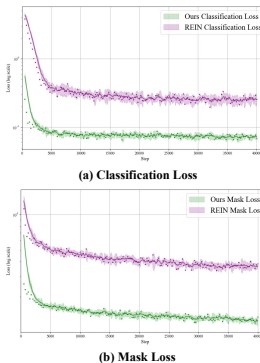

(a) Classification Loss

(b) Mask Loss

Figure 7: Comparison of Train Loss for Baseline and CMPE Models.

**Pronounced Attention Focus.** Figure 5 further confirm that GeoText Query steer the model toward geometry-sensitive regions, while CMPE strengthens gradients on

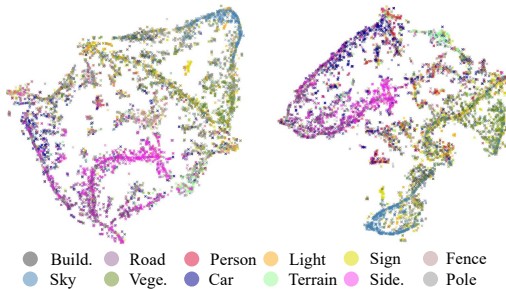

Build. Road Person Light Sign Fence
Sky Vege. Car Terrain Side. Pole

Figure 6: t-SNE embeddings of the feature space. **Left**: original source-domain dataset. **Right**: our **Vireo** after adaptation on *Cityscapes → ACDC + BDD 100k + Mapillary*. Each point is coloured by its semantic class.

| Configurations | Snow | Night | Fog | Rain | BDD. | Map. |
|---|---|---|---|---|---|---|
| REIN [36] | 70.6 | 55.9 | 79.5 | 72.5 | 63.5 | 74.0 |
| + concat $f_i^d$ | 70.8 | 56.1 | 79.4 | 72.8 | 63.6 | 74.2 |
| + Prompt Depth Anything | 70.4 | 55.5 | 79.7 | 72.0 | 63.8 | 73.9 |
| + Depth Anything V2 | 71.5 | 56.7 | 80.5 | 73.3 | 64.4 | 74.5 |
| + DA + AO | 72.2 | 57.4 | 80.9 | 74.2 | 65.1 | 75.0 |
| + DOV-VEH | 70.9 | 56.2 | 79.8 | 72.8 | 63.7 | 74.2 |
| + CMPE | 71.6 | 56.9 | 80.2 | 73.4 | 64.1 | 74.6 |
| + GeoText Query | 74.0 | 58.4 | 81.1 | 74.8 | 65.3 | 75.3 |
| **Vireo** | 76.2 | 60.6 | 82.3 | 76.3 | 66.7 | 76.0 |

Table 5: Ablation studies on component configurations of the proposed **Vireo** under the *Citys.→ACDC with Snow, Night, Fog, Rain + BDD. + Map.* generalization setting. Top three results are highlighted as best , second , and third (%).

**CLIP** (ViT-L) [55]

| Fine-tune | Trainable Params* | mIoU | | | Avg. |
|---|---|---|---|---|---|
| | | Citys | BDD | Map | |
| Full | 304.15M | 51.3 | 47.6 | 54.3 | 51.1 |
| Freeze | 0.00M | 53.7 | 48.7 | 55.0 | 52.4 |
| REIN | 2.99M | 57.1 | 54.7 | 60.5 | 57.4 |
| FADA | 11.65M | 58.7 | 55.8 | 62.1 | 58.9 |
| **Vireo (Ours)** | 3.78M | **60.5** | **57.5** | **64.1** | **60.7** |

**SAM** (Huge) [56]

| Fine-tune | Trainable Params* | mIoU | | | Avg. |
|---|---|---|---|---|---|
| | | Citys | BDD | Map | |
| Full | 632.18M | 57.6 | 51.7 | 61.5 | 56.9 |
| Freeze | 0.00M | 57.0 | 47.1 | 58.4 | 54.2 |
| REIN | 4.51M | 59.6 | 52.0 | 62.1 | 57.9 |
| FADA | 16.59M | 61.0 | 53.2 | 63.4 | 60.0 |
| **Vireo (Ours)** | 5.30M | **64.5** | **59.0** | **66.0** | **63.2** |

**EVA02** (Large) [46, 47]

| Fine-tune | Trainable Params* | mIoU | | | Avg. |
|---|---|---|---|---|---|
| | | Citys | BDD | Map | |
| Full | 304.24M | 62.1 | 56.2 | 64.6 | 60.9 |
| Freeze | 0.00M | 56.5 | 53.6 | 58.6 | 56.2 |
| REIN | 2.99M | 65.3 | 60.5 | 64.9 | 63.6 |
| FADA | 11.65M | 66.7 | 61.9 | 66.1 | 64.9 |
| **Vireo (Ours)** | 3.78M | **68.5** | **62.1** | **67.4** | **66.0** |

**DINOv2** (Large) [54]

| Fine-tune | Trainable Params* | mIoU | | | Avg. |
|---|---|---|---|---|---|
| | | Citys | BDD | Map | |
| Full | 304.20M | 63.7 | 57.4 | 64.2 | 61.7 |
| Freeze | 0.00M | 63.3 | 56.1 | 63.9 | 61.1 |
| REIN | 2.99M | 66.4 | 60.4 | 66.1 | 64.3 |
| FADA | 11.65M | 68.2 | 62.0 | 68.1 | 66.1 |
| **Vireo (Ours)** | 3.78M | **70.7** | **62.9** | **69.6** | **67.7** |

Table 6: Performance comparison of **Vireo** across multiple VFMs under the *GTA5 → Citys. + BDD. + Map.* setting. *Trainable parameters in backbone. Top three results are best , second , third . (%)

foreground masks; compared with baselines, Vireo's focus is tighter on scene structure and semantic boundaries, explaining its consistent advantage in cross-domain and open-vocabulary settings.

**t-SNE Visualization of DGSS Features.** The feature distributions of the original dataset and our method are visualized in Figure 6, revealing the superiority of our learned features in forming well-separated semantic clusters. This demonstrates the effectiveness of our domain-generalized visual-textual alignment in structuring the open-vocabulary semantic space.

## 5 Conclusion

This study introduces Vireo, the first single-stage framework that unifies open-vocabulary recognition and domain-generalised semantic segmentation. By integrating frozen visual foundation models, depth-aware geometry and three core modules—GeoText Query, Coarse Mask Prior Embedding and the DOV Vector Embedding Head—it converges faster, concentrates attention on scene structure and outperforms state-of-the-art methods across multiple benchmarks, demonstrating the power of combining textual cues with geometric priors for robust pixel-level perception.

**Limitations** Vireo assumes a reliable RGB camera, but in rare cases (e.g., occlusion, glare, hardware failure), the stream can be lost, impairing perception. Future work will explore multi-source setups—automatically switching to lidar, radar, or event-based cameras when RGB fails—to keep segmentation robust in all conditions.

## Acknowledgements

This work was supported in part by the Natural Science Foundation of Xiamen, China, under Grants 3502Z202373036 and 3502Z202371019; in part by the National Natural Science Foundation of China under Grant 42371457; in part by the Natural Science Foundation of Fujian Province, China, under Grants 2025J01345 and 2023J01803; by the 2025 University of Glasgow Early Career Reward for Excellence on "*Developing Vision–Language Models Enhanced by Geospatial Intelligence*" and the 2025 University of Glasgow Early Career Mobility Scheme (ECMS) Award; and by the Program of China Scholarship Council under Grant No. 202506380088.

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
