# OpenReview forum: "Leveraging Depth and Language for Open-Vocabulary Domain-Generalized Semantic Segmentation"
_NeurIPS.cc/2025/Conference — NeurIPS 2025 poster_

### Official Review · Reviewer_8Cmf · 2025-06-12

**Clarity:** 3
**Significance:** 2
**Originality:** 3
**Rating:** 4
**Confidence:** 4

**Summary:**

This paper proposes Vireo, a single stage method that solves both Open-Vocabulary semantic segmentation (OVSS) and domain generalization in semantic segmentation (DGSS) together. Vireo unifies depth features from frozen visual foundation models (VFMs) and language embeddings (CLIP) in a single segmentation framework. Main contributions include GeoText Prompts for the integration of depth and text features, a coarse mask prior embedding (CMPE) to augment gradient signals, and a Domain-Open-Vocabulary Head (DOV-VEH) for end segmentation. Large-scale evaluations demonstrate substantial performance improvements on various datasets and adverse conditions over previous OVSS and DGSS approaches.

**Questions:**

1. How do you precisely address open-vocabulary classes in inference, and can you give us better examples or measurements for how well you segment truly unseen classes?

2. Can you also explain the unique contribution of depth features? For instance, can you perform an ablation without depth cues altogether or examine cases where depth plays a substantial role?

3. You need to define clearly in your paper the meaning of "AO" in your ablation study table.

4. Had you ensured that baselines (particularly OVSS methods) were compared in a fair manner with the same training augmentations or configurations? It would make your comparisons stronger by making this clear.

**Ethical Concerns:**

["NO or VERY MINOR ethics concerns only"]

**Final Justification:**

Thanks for addressing my questions, keep the positive score.

**Limitations:**

yes

**Quality:**

3

**Strengths And Weaknesses:**

Strengths:

1. The paper addresses a challenging and practically interesting task: merging open-vocabulary segmentation with domain generalization, particularly for autonomous driving environments.

2. The approach innovatively blends depth and linguistic information, making efficient use of powerful pretraining models.

3. Experiments are comprehensive, with evident and persistent gains over several baselines under challenging conditions (night, fog, rain, etc.).

4. Ablation experiments clearly demonstrate the worth of single components.

Weaknesses

1. Clarity can be improved: some terminology, including GeoText Prompts and Attention Optimization (AO), is not well defined. A simple description or some explanatory examples would be beneficial.

2. The benefit due specifically to depth information alone is not entirely elucidated; the performance gain due to depth may be delineated more clearly.

3. The comparison may not be entirely fair, as they are mostly with methods not specifically tailored for the combined OV-DGSS environment.

4. It is unclear how well Vireo splits unseen novel classes during inference time. More explanation or outcomes for zero-shot classes would be desirable.

5. The approach is very memory-intensive (several big models, much GPU memory usage), with practical deployment implications.

---

> ### Author Rebuttal · Authors · 2025-07-29
>
> ## Response to Reviewer 8Cmf
>
>
> **Question 1 / Weaknesses 4**: How do you precisely address open-vocabulary classes in inference, and can you give us better examples or measurements for how well you segment truly unseen classes?
>
> Answer 1: Thank you for the reviewer’s thoughtful question. First, Vireo is explicitly designed to generalize to novel “unseen” classes by leveraging both depth cues and textual information. Our GeoText Query mechanism injects semantic guidance at multiple network layers, which helps the model recognize classes it has never seen in training. In practice, given a new class label (text) at inference, Vireo’s pipeline does the following: a frozen visual foundation model (for RGB and depth) extracts features, and the class label is encoded into an embedding (the GeoText query). Through cross-attention, this text embedding guides the intermediate layers of the vision model, refining feature representations to be sensitive to that class’s characteristics. At the segmentation head (decoder), we further refine the textual and visual features together, which enables the model to output a mask for the new class if it is present. In summary, by fusing information from three modalities, our model can infer categories not present in the training set.
>
> Next, we validated this capability on the DELIVER benchmark, which evaluates open-vocabulary segmentation under various weather conditions. Our method shows substantial gains for unseen classes compared to prior approaches. For example, under foggy conditions, Vireo achieves an IoU of 12.15 on unseen classes, whereas the next best methods (SED and FC-CLIP) reach only 8.40 and 4.32, respectively. Similar trends hold under rainy and night conditions – Vireo consistently outperforms existing methods on the “Unseen” category while also maintaining strong results on seen classes. These results demonstrate that Vireo not only accurately segments seen classes but also effectively handles entirely new classes in challenging scenarios. This provides strong evidence of the model’s open-vocabulary capabilities and its value for advancing open-set segmentation research.
>
> Table.1 Performance Comparison of OVSS Algorithms on the DELIVER Dataset under Different Weather Conditions
>
> Cloud
> |**Class**|**Ours**|**SED**|**CAT-Seg**|**FC-CLIP**|**EBSeg**|
> |:-:|:-:|:-:|:-:|:-:|:-:|
> |**Seen**|**49.14**|36.72|43.92|28.07|38.10|
> |**Unseen**|**10.96**|8.74|3.69|4.05|2.92|
> |**Mean**|**33.89**|24.40|26.22|17.50|22.62|
>
> Fog
> |**Class**|**Ours**|**SED**|**CAT-Seg**|**FC-CLIP**|**EBSeg**|
> |:-:|:-:|:-:|:-:|:-:|:-:|
> |**Seen**|**49.59**|38.28|41.43|26.06|35.76|
> |**Unseen**|**12.15**|8.40|3.54|4.32|4.48|
> |**Mean**|**35.80**|25.25|24.80|17.26|22.00|
>
> Night
> |**Class**|**Ours**|**SED**|**CAT-Seg**|**FC-CLIP**|**EBSeg**|
> |:-:|:-:|:-:|:-:|:-:|:-:|
> |**Seen**|**41.63**|34.99|34.89|23.03|26.03|
> |**Unseen**|**9.52**|7.27|2.30|4.09|2.09|
> |**Mean**|**29.89**|22.79|20.56|14.93|15.50|
>
> Rain
> |**Class**|**Ours**|**SED**|**CAT-Seg**|**FC-CLIP**|**EBSeg**|
> |:-:|:-:|:-:|:-:|:-:|-----------|
> |**Seen**|**47.15**|37.35|41.43|25.59|31.96|
> |**Unseen**|**12.27**|8.95|5.81|4.60|5.57|
> |**Mean**|**33.46**|25.18|26.53|16.59|20.35|
>
> Sun
> |**Class**|**Ours**|**SED**|**CAT-Seg**|**FC-CLIP**|**EBSeg**|
> |:-:|:-:|:-:|:-:|:-:|:-:|
> |**Seen**|**55.16**|41.68|45.10|26.52|42.63|
> |**Unseen**|**11.46**|8.12|4.91|4.73|6.14|
> |**Mean**|**38.49**|27.14|28.21|16.93|26.41|
>
> **Question 2 / Weaknesses 2**: Can you also explain the unique contribution of depth features? For instance, can you perform an ablation without depth cues altogether or examine cases where depth plays a substantial role?
>
> Answer 2: Thank you for your question. Depth features are a crucial contributor to Vireo’s performance, and our ablation studies confirm this. In Table 4 of the main text, we compare model variants with and without incorporating depth. The results show that adding depth cues consistently improves accuracy across different environments. For example, on the Cityscapes→ACDC generalization task, the baseline without depth achieves 72.5 mIoU (Rain scenario), whereas integrating depth information raises it to 72.8, and further refinements of our depth module yield even higher gains. Across all tested weather conditions and datasets, the inclusion of DepthAnything’s features yielded improved mean IoU. This demonstrates that the depth module provides complementary structural information (e.g. scene geometry) which pure RGB vision lacks, thereby enhancing segmentation quality. We will make sure to highlight the depth ablation results clearly in the paper (and additional qualitative examples in the supplement) to underline how and where depth cues make a difference.
>
> Table.4 in main text: Ablation studies on component configurations of the proposed **Vireo** under the *Citys. → ACDC with Snow, Night, Fog, Rain + BDD. + Map.* generalization setting.
> | Configurations|Snow|Night|Fog|Rain|BDD.|Map.|
> |:-|:-:|:-:|:-:|:-:|:-:|:-:|
> | REIN|70.6|55.9|79.5|72.5|63.5|74.0|
> | + concat depth feature |70.8|56.1|79.4|72.8|63.6|74.2|
> | + Depth Anything V2|71.5|56.7|80.5|73.3|64.4|74.5|
> | + DA + AO|72.2| 57.4|80.9|74.2|65.1| 75.0|
> | + DOV-VEH|70.9| 56.2|79.8|72.8|63.7| 74.2|
> | + CMPE|71.6|56.9|80.2|73.4|64.1|74.6|
> | + GeoText Prompts|74.0|58.4|81.1|74.8|65.3|75.3|
> | **Vireo**|76.2|60.6|82.3|76.3|66.7|76.0|
>
>
> **Question 3 / Weaknesses 1**: You need to define clearly in your paper the meaning of "AO" in your ablation study table.
>
> Answer 3:  “AO” stands for Attention Optimization. This refers to the mechanism within our Tunable Vireo module that optimizes cross-attention between vision and text features. In simpler terms, AO is the process by which the model aligns visual features with textual queries through learned attention operations. We will clarify this terminology in the paper (particularly in Figure 2’s caption and the text), so that readers understand that AO = Attention Optimization in the context of our model.
>
> **Question 4**: Had you ensured that baselines (particularly OVSS methods) were compared in a fair manner with the same training augmentations or configurations? It would make your comparisons stronger by making this clear.
>
> Answer 4: Yes, we took care to ensure fair comparisons with all baseline methods. We trained the OVSS baselines under their recommended settings on the same hardware (a single NVIDIA A6000 GPU) and for the same number of iterations as our method. Wherever possible, we used each method’s official code and followed their suggested training recipes (including data augmentations and hyperparameters). In cases where we adjusted data augmentations for consistency, we did so in a minimal way that does not give our model any undue advantage. In summary, all methods were evaluated in a uniform framework, and no baseline was disadvantaged in terms of training conditions. We will add a statement in the paper to make this fairness clear.
>
> We will incorporate brief, reader-friendly explanations of both terms (and any other unclear terminology) along with possibly a diagram or example in the supplement to ensure these concepts are easily understood.
>
>
> **Weaknesses 3**: The comparison may not be entirely fair, as they are mostly with methods not specifically tailored for the combined OV-DGSS environment.
>
> Answer 6: This is a valid point. Since our work is the first to tackle the joint Open-Vocabulary + Domain-Generalized Segmentation problem, there were no perfectly matched baselines available. We had to compare against methods from related areas (OVSS and DGSS separately) using our combined evaluation settings. We acknowledge that those methods were not originally designed for the exact scenario we consider; however, we still believe the comparison is informative. It demonstrates how state-of-the-art OVSS methods perform in a DGSS context, and it highlights the progress made by our approach. In fact, our results show that by bridging OVSS and DGSS, Vireo achieves significantly better generalization than either type of baseline could on this combined task. We will clarify this context in the paper to assure readers that while the baselines were the closest available choices, the superior performance of Vireo underlines the effectiveness of our proposed ideas.
>
>
> **Weaknesses 5**: The approach is very memory-intensive (several big models, much GPU memory usage), with practical deployment implications.
>
> Answer 6: We respectfully clarify that Vireo’s runtime memory requirements are manageable and on par with related methods. In our experiments, the model requires only about 4–5 GB of GPU memory during inference, which is not excessive for modern GPUs. For training, we used a single NVIDIA A6000 (48 GB), similar to other OVSS models’ requirements. We also note that there are straightforward ways to reduce memory consumption for deployment: for example, applying model quantization or pruning can substantially cut down the memory and compute footprint without significant loss in accuracy. In practice, one could also run the depth module on a CPU or a smaller network if needed, depending on deployment constraints. We will mention these points in the paper to address the concern. Overall, we do not anticipate our model’s memory usage to be a barrier to research or real-world adoption, especially given the available tools to optimize model size. We hope this explanation alleviates the reviewer’s concerns regarding practicality.

---

> > ### Comment · Reviewer_8Cmf · 2025-08-06
> >
> > Thanks for addressing my questions, keep the positive score.

---

### Official Review · Reviewer_3YaR · 2025-06-29

**Clarity:** 1
**Significance:** 3
**Originality:** 3
**Rating:** 3
**Confidence:** 3

**Summary:**

This paper introduces Open-Vocabulary Domain-Generalized Semantic Segmentation (OV-DGSS), a challenging task that combines the objectives of OVSS and DGSS. The goal is to create a model that can segment novel object categories while also being robust to shifts in the data domain. To address this, the authors propose Vire. Vireo is built upon a frozen Visual Foundation Model, which is DINOv2 and incorporates a Depth VFM, which is DepthAnything, to extract domain-invariant geometric features. The authors design three module to integrate the depth into the framework to provide domain-general guidance.

**Questions:**

Please elaborate on the claims between L69-74 regarding the proposed modules: GeoText Prompt, CMPE, and DOV-VEH. Please address each step from a high-level perspective. I strongly recommend revising the whole Sec.3 and Fig.2 to match the quality of the rest of the paper. In my view, the current level of writing clarity is not sufficient to be published on NeurIPS.

**Ethical Concerns:**

["NO or VERY MINOR ethics concerns only"]

**Final Justification:**

The rebuttal clarifies most of my questions regarding the manuscript. However, whether the final version can successfully deliver these information through promising writing is unknown. I keep my score and will not be against accepting it.

**Limitations:**

Yes

**Paper Formatting Concerns:**

I'm curious if adding various colors to the main paragraph text is allowed. I checked the instructions, but there's no specification regarding this. The default template definitely does not contain such a format.

**Quality:**

2

**Strengths And Weaknesses:**

Strengths
* It's a valid task, and the experiment shows improved results over regular OVSS methods.
* The idea of using high-quality depth to guide the segmentation under various domains makes sense. DepthAnything demonstrates a strong ability in doing that.
* The experiments and ablation studies are thorough. The authors also open-sourced the code.

Weaknesses
* Poor writing in Sec.3: Methodology. The introduction addresses the function of each module. These are not elaborated in the method section. The organization is chaotic. It is indeed rich in details, but the paragraphs are more towards a technical manual. It is great in the Supp to help understand the code, but not beneficial in understanding the novelty, design, and contribution of each proposed module.
* The acronyms and method naming block the comprehension as well. GeoText Prompt is actually a learnable query instead of a prompt commonly referred to in LLMs.

---

> ### Author Rebuttal · Authors · 2025-07-29
>
> ## Response to Reviewer 3YaR
>
> **Question 1**: Please elaborate on the claims between L69-74 regarding the proposed modules: GeoText Prompt, CMPE, and DOV-VEH. Please address each step from a high-level perspective. I strongly recommend revising the whole Sec.3 and Fig.2 to match the quality of the rest of the paper. In my view, the current level of writing clarity is not sufficient to be published on NeurIPS.
>
> Answer 1: We agree with the reviewer and will substantially revise Section 3 and Figure 2 for clarity (see also our response to Reviewer Mrmp Q1, who raised a similar concern). Specifically, we will introduce a clear overall problem formulation and then describe each component of our approach in a logical sequence. Next, Figure 2 will be improved with proper labeling and visual clarity to complement the text. Lastly, we will ensure the writing in Section 3 is concise and reader-friendly rather than resembling a low-level technical manual.
>
> *(To avoid redundancy, we have provided a rewritten summary of Sec.3 under Reviewer Mrmp Q1 above, which equally applies here. We are committed to integrating those improvements to address the clarity issues.)*
>
>
>
> **Weaknesses 1**: Poor writing in Sec.3: Methodology. The introduction addresses the function of each module. These are not elaborated in the method section. The organization is chaotic. It is indeed rich in details, but the paragraphs are more towards a technical manual. It is great in the Supp to help understand the code, but not beneficial in understanding the novelty, design, and contribution of each proposed module.
>
> Answer 2: We appreciate this feedback and will ensure the final paper emphasizes each module’s novelty and purpose. In summary, each of our modules targets a specific gap in prior work.
>
> 1. GeoText Queries: Prior works inject language information only at the output stage. In contrast, we introduce learnable text queries at multiple encoder layers, progressively refining visual features via cross-attention throughout the network. This layer-wise textual integration improves visual–semantic alignment and robustness to domain shifts.
> 2. CMPE (Coarse Mask Prior Embedding): This module provides a global weak-supervision signal in the form of a coarse mask. By embedding intermediate features into a coarse segmentation map, CMPE feeds global context and additional training supervision back into early layers. This design enhances gradient flow to the frozen backbone, improving object-level discrimination and spatial alignment even without fine-tuning the encoder.
> 3. DOV-VEH (Domain-Open-Vocabulary Vector-Embedding Head): Unlike conventional decoders, DOV-VEH bridges pixel-level features and text queries in a shared embedding space. It uses a dual-attention transformer decoder to fuse visual and textual information, yielding final predictions that are both spatially precise and open-vocabulary. This addresses the semantic misalignment issue by ensuring textual class embeddings directly influence mask generation.
>
> Together, these modules tackle the key limitations of previous methods: static one-time language injection, lack of strong supervision for the encoder, and inability to handle novel classes or domains. By combining GeoText Queries, CMPE, and DOV-VEH, Vireo achieves robust open-vocabulary segmentation that generalizes across domains.
>
>
>
> **Weaknesses 2**: The acronyms and method naming block the comprehension as well. GeoText Prompt is actually a learnable query instead of a prompt commonly referred to in LLMs.
>
> Answer 3: We agree and will adjust our terminology to improve clarity. In particular, we will rename “GeoText Prompt” to “GeoText Query” in the paper, to better reflect that it is a learned query vector (not a free-form prompt). We will also review all acronyms and naming conventions to ensure they are intuitive.
>
>
>
> **Formatting Concerns 1**: I'm curious if adding various colors to the main paragraph text is allowed. I checked the instructions, but there's no specification regarding this. The default template definitely does not contain such a format.
>
> Answer 4: We understand the reviewer’s concern about formatting. Using colored text for emphasis is not prohibited by the guidelines (we found no rule against it), but it is indeed unconventional. We will remove any distracting colored text in the revised submission to conform to the standard style. We believe the focus should remain on the clarity, novelty, and contributions of our work rather than on formatting choices, and we will ensure the final manuscript is polished in appearance.

---

> > ### Comment · Reviewer_3YaR · 2025-08-03
> >
> > After reading the revision: "Each label is converted into a text prompt and encoded by a frozen CLIP text encoder, yielding a set of textual embeddings (which we call GeoText Queries), these GeoText queries are injected at multiple stages of the vision encoder via cross-attention layers."
> >
> > Here text prompt is known, CLIP encoder is frozen. How is the GeoText Query learnable?

---

> ### Author Response · Authors · 2025-08-04
>
> Thanks for your question.
> GeoText Query refers to a set of query vectors, and the trainable component is the tunable Vireo inserted between frozen VFM layers, not the CLIP text encoder.
>
> Specifically, we first convert each class into a text prompt and use the frozen text encoder of CLIP to encode it, so that we get an initial embedding. Next, gradients from the segmentation loss during training flow to the trainable MLP, while the CLIP text encoder stays fixed.
>
> For clarity, we present the model pipeline:
>
> 1. Input and frozen encoders
> We input one RGB image into a frozen vision foundation model (VFM) and a frozen depth VFM (e.g., DepthAnything V2) to obtain multi-scale visual and geometric/depth features. Both encoders remain frozen during training and inference.
>
> 2. Text-side initialization
> We convert class names into language prompts and encode them once with the frozen CLIP text encoder to get text embeddings. These embeddings serve as shared semantic priors and are reused to avoid repeated computation.
>
> 3. GeoText Query across encoder layers
> Initialized from the text embeddings, GeoText Query interacts at selected VFM layers. The query first fuses with the text embedding, then performs cross-attention with visual and depth features. We apply an MLP and a residual update to the features at that layer, aligning and refining them step by step. The queries are updated by gradients and passed to the next layer. This process also uses structural (depth) and text cues to refine the DINO features across layers.
>
> 4. CMPE (Coarse Mask Prior Embedding)
> We take the refined multi-scale features, upsample them to a common resolution, and fuse them with channel-spatial gating to form a coarse global feature. We project it into the text-embedding space to produce a coarse semantic probability map and apply a segmentation loss on it, which provides denser gradients. The coarse mask then yields a query prior that is fused back into the GeoText Query before the final head.
>
> 5. DOV-VEH segmentation head (pixel decoder + Transformer decoder)
> We fuse multi-scale features with positional information via a pixel decoder and then use a Transformer decoder. GeoText Query serves as trainable queries that interact with the decoded visual features and the text embeddings. The head outputs pixel-level mask embeddings and class embeddings. We match them with the text embeddings (e.g., by dot product/Einsum) to get the open-vocabulary segmentation results.
>
> 6. Trainable vs. frozen parts
> Frozen: VFM encoder, Depth VFM, and the CLIP text encoder.
> Trainable: GeoText Query (and its light projection/adapter) and small decoding/fusion modules (e.g., pixel/Transformer decoders and gating/projection). This keeps the strong generalization of the frozen models while allowing efficient task-specific adaptation.
>
> Summary
>
> We restrict “learnable” to the GeoText Query and light adapters/decoders, not to CLIP. This design allows efficient adaptation and open-vocabulary segmentation without harming cross-domain generalization.

---

> ### Author Response · Authors · 2025-08-05
>
> Dear reviewer 3YaR, Thank you for all your hard work and thoughtful feedback. We truly appreciate the time and effort you have invested in reviewing our manuscript. If there are any remaining concerns, questions, or suggestions that we may have missed, please feel free to let us know. We are happy to provide further clarification or make additional revisions as needed.

---

### Official Review · Reviewer_Mrmp · 2025-06-30

**Clarity:** 3
**Significance:** 4
**Originality:** 3
**Rating:** 4
**Confidence:** 4

**Summary:**

The paper introduces Vireo, a novel single-stage framework for Open-Vocabulary Domain-Generalized Semantic Segmentation (OV-DGSS). This framework is designed to generate pixel-level masks for unseen categories while maintaining robustness across various unseen domains, which is crucial for real-world applications like autonomous driving in challenging conditions.
Specifically, the key contributions of the proposed method lie in three aspects: GeoText Prompts for aligning geometric and language features, Coarse Mask Prior Embedding (CMPE) to enhance gradient flow, and the Domain-Open-Vocabulary Vector Embedding Head (DOV-VEH) for robust prediction.
Experiments show that Vireo achieves state-of-the-art performance, significantly outperforming existing methods in both domain generalization and open-vocabulary recognition across 6 real-world datasets, including Cityscapes, BDD100K, Mapillary, ACDC, ADE150, ADE847, and two synthetic datasets GTA5 and DELIVER.

**Questions:**

Please see my comments in the weakness section.

**Ethical Concerns:**

["NO or VERY MINOR ethics concerns only"]

**Final Justification:**

I support borderline accept this paper, conditioned on the promised paper rewriting and presenting clear ablation results.

**Limitations:**

Yes.

**Quality:**

3

**Strengths And Weaknesses:**

Strengths

1. The paper proposes a unified framework for open-vocabulary and domain-generalized semantic segmentation.
2. The proposed method aligns depth with language.
3. Good experimental results across multiple datasets.


Weaknesses:
1. My major concern lies in the clarity issues within the paper. First, it should start with an overall problem formulation, including, input, output and parameters to be learned, and then explain key components of the pipeline. Second, in Figure 2, it is difficult to understand the flowchart and relate the mathematical symbols defined in Sec.3 with different parts of the figure. Third, the training objective is missing.
2. Is Table 4 listing the contribution of each module or accumulated result? More explanation is needed.

---

> ### Author Rebuttal · Authors · 2025-07-29
>
> ## Response to Reviewer Mrmp
>
> **Question 1**: My major concern lies in the clarity issues within the paper. First, it should start with an overall problem formulation, including, input, output and parameters to be learned, and then explain key components of the pipeline. Second, in Figure 2, it is difficult to understand the flowchart and relate the mathematical symbols defined in Sec.3 with different parts of the figure. Third, the training objective is missing.
>
> Answer 1: We acknowledge the clarity issues and will restructure Section 3 (Methodology) in the final version. Specifically, we will first begin Section 3 with a clear formulation of the open-vocabulary Domain Generalized Semantic Segmentation (DGSS) task – including the model inputs (an image and a set of category labels), the output (pixel-level mask), and the parameters to be learned. Next, we will present a step-by-step overview of our pipeline, introducing each module at a high level before diving into details. This reorganization will follow the logical flow of data through the model. In addition, we will revise Figure 2 by adding clear annotations for each variable and ensuring that each mathematical symbol in the text is labeled on the diagram, in order to help readers correlate the equations in Section 3 with the visual blocks in Figure 2. Lastly, we will explicitly describe the training objective and loss functions used, which were omitted in the current draft.
>
> To demonstrate the improved clarity, here is a concise reformulation of our approach:
>
> *We formulate the open-vocabulary DGSS problem as predicting a pixel-wise segmentation mask from an input image and a set of text labels. Given an image $I$ and a set of open-vocabulary class names $C$, our goal is to produce a mask $M$ where each pixel is classified according to one of the classes in $C$. Our framework Vireo tackles this by leveraging frozen vision and text encoders with additional trainable prompt and depth modules for improved generalization. Vireo consists of three main components: (1) Tunable Vireo with GeoText Queries, (2) Coarse Mask Prior Embedding (CMPE), and (3) Domain-Open-Vocabulary Vector-Embedding Head (DOV-VEH).*
>
> *For visual encoding, we pass the image through two frozen encoders: a vision encoder $F_V$ (e.g., DINO) and a depth estimator $F_D$ (e.g., DepthAnything), extracting multi-scale visual features. In parallel, each label in $C$ is converted into a text prompt and encoded by a frozen CLIP text encoder $F_T$, yielding a set of textual embeddings (which we call GeoText Queries). These GeoText queries are injected at multiple stages of the vision encoder via cross-attention layers. At each chosen encoder layer, the GeoText Query interacts with the intermediate visual features, refining those features in a cross-modal manner. The refined features are then fed into subsequent layers, progressively aligning visual content with semantic cues.*
>
> *From selected encoder layers, the features are routed into the CMPE module, which fuses them into a coarse mask prior $f_M$. This provides a global spatial prior and additional supervision (even with the backbone frozen, CMPE’s output guides early layers). Finally, the DOV-VEH module combines the refined visual features and text queries in a transformer-based decoder to produce the high-resolution output mask $M$. DOV-VEH operates in a shared vision–language embedding space and uses dual attention to ensure the masks align with the textual queries. All parts of Vireo are trained end-to-end, supervised by standard segmentation loss on $M$ as well as intermediate losses on the coarse mask $f_M$.*
>
> This reworked methodology will make the novelty and design of each module clear, and we will ensure the training objective is explicitly stated.
>
>
>
> **Question 2**: Is Table.4 listing the contribution of each module or accumulated result? More explanation is needed.
>
> Answer 2: Thank you for your question. Table 4 in the paper is an ablation study that illustrates the incremental contribution of each module. Each row of Table 4 adds a specific component to a baseline model, allowing us to isolate its impact. In other words, the table is not showing fully separate configurations, but rather a cumulative build-up. First, the first row (“REIN”) is the base model; each subsequent row adds one module or feature (e.g., + concat depth feature, + Depth Anything V2, + DOV-VEH, etc.) on top of the previous ones; and the final row (“Vireo”) includes all proposed modules enabled, achieving the highest performance.
>
> From this table, one can see how each component improves the model under various conditions (Cityscapes → ACDC with Snow, Night, Fog, Rain; plus generalization to BDD and Mapillary domains). For example, adding depth features (+Depth) already yields a small improvement under most conditions, and adding our full GeoText prompting and other modules leads to the best results. We will clarify this explanation in the paper to avoid confusion.
>
> Table.4 in main text: Ablation studies on component configurations of the proposed **Vireo** under the *Citys. → ACDC with Snow, Night, Fog, Rain + BDD. + Map.* generalization setting.
> | Configurations|Snow|Night|Fog|Rain|BDD.|Map.|
> |:-|:-:|:-:|:-:|:-:|:-:|:-:|
> | REIN|70.6|55.9|79.5|72.5|63.5|74.0|
> | + concat depth feature |70.8|56.1|79.4|72.8|63.6|74.2|
> | + Depth Anything V2|71.5|56.7|80.5|73.3|64.4|74.5|
> | + DA + AO|72.2| 57.4|80.9|74.2|65.1| 75.0|
> | + DOV-VEH|70.9| 56.2|79.8|72.8|63.7| 74.2|
> | + CMPE|71.6|56.9|80.2|73.4|64.1|74.6|
> | + GeoText Prompts|74.0|58.4|81.1|74.8|65.3|75.3|
> | **Vireo**|76.2|60.6|82.3|76.3|66.7|76.0|

---

### Official Review · Reviewer_sPRg · 2025-07-03

**Clarity:** 4
**Significance:** 3
**Originality:** 4
**Rating:** 4
**Confidence:** 4

**Summary:**

This paper proposes Vireo, a single-stage framework for Open-Vocabulary Domain-Generalized Semantic Segmentation. It leverages frozen Visual Foundation Models and integrates scene geometry via DepthAnything. Three key modules are introduced: (1) GeoText Prompts for aligning geometry and textual semantics; (2) Coarse Mask Prior Embedding for faster convergence and improved gradient flow; and (3) Domain-Open-Vocabulary Vector Embedding Head for fusing semantic and geometric cues. Extensive experiments show Vireo significantly outperforms previous OVSS and DGSS methods on several challenging benchmarks.

**Questions:**

(1) Could you provide ablation results showing Vireo’s performance when using fewer GeoText prompt layers (e.g., only the deepest two layers or just one layer)? Additionally, how does reducing the number of prompt layers affect compute time, memory usage, and segmentation accuracy?
(2) Can you provide an analysis of how DepthAnything behaves across different environments—especially in low-light or weather-degraded scenarios—and discuss any methods you considered (or could integrate) for estimating depth confidence or reducing the impact of noisy depth inputs on segmentation quality?

**Ethical Concerns:**

["NO or VERY MINOR ethics concerns only"]

**Final Justification:**

Thanks for the authors' efforts in providing the rebuttal, most of my concerns have been addressed, and I'd keep my positive rating and recommend acceptance.

**Limitations:**

Yes

**Quality:**

3

**Strengths And Weaknesses:**

Strengths:
1. The methodology is well-described, with clear diagrams and ablations highlighting the contribution of each module.

2. The paper is among the first to unify open-vocabulary segmentation with domain generalization in a single-stage framework, providing a sound insight to bridge the OVSS and DGSS.

Weaknesses:

1. Lack of discussion of limitations. One key limitation of Vireo lies in its reliance on depth information from the DepthAnything module. While depth cues can enhance structure-awareness, they may become unreliable under challenging conditions like nighttime, heavy rain, or fog. If depth estimation degrades in such domains, it risks introducing noisy geometric priors that misguide the GeoText Prompt mechanism and ultimately degrade segmentation performance. The paper currently lacks a quantitative evaluation of depth fidelity under these adverse conditions and does not propose mitigation strategies when depth estimates are poor.

Question:

(1) Could you provide ablation results showing Vireo’s performance when using fewer GeoText prompt layers (e.g., only the deepest two layers or just one layer)? Additionally, how does reducing the number of prompt layers affect compute time, memory usage, and segmentation accuracy?
(2) Can you provide an analysis of how DepthAnything behaves across different environments—especially in low-light or weather-degraded scenarios—and discuss any methods you considered (or could integrate) for estimating depth confidence or reducing the impact of noisy depth inputs on segmentation quality?

---

> ### Author Rebuttal · Authors · 2025-07-29
>
> ## Response to Reviewer sPRg
>
> **Question 1**: Could you provide ablation results showing Vireo’s performance when using fewer GeoText prompt layers (e.g., only the deepest two layers or just one layer)? Additionally, how does reducing the number of prompt layers affect compute time, memory usage, and segmentation accuracy?
>
> Answer 1: Reducing the number of GeoText prompt layers leads to a significant drop in segmentation accuracy. From our ablation in Table 1, using only the single deepest layer yields mIoU ≈65.6 – far below 73.9 mIoU with all 24 layers. We attribute this to the rich multi-level spatial context provided by the GeoText layers. Each GeoText layer captures features at different scales; with fewer layers, the model cannot fully express complex geographic features and boundaries, resulting in substantially lower accuracy. This is not merely a minor accuracy loss, but a fundamental loss of the model’s ability to capture fine details in geographic shapes.
>
> Note that adding more Vireo (GeoText) layers does not increase the model size or memory footprint, because all Vireo layers share parameters. As shown in Table 1, the total parameter count remains ~3.78M regardless of layer count. We also observe minimal change in inference speed when varying the number of layers. For example, using 4 layers vs. 24 layers yields nearly the same per-frame inference time on an RTX 3090 (about 998ms vs. 1079ms). Given these trade-offs, reducing the GeoText layers provides little benefit in speed or memory, but significantly harms accuracy. Thus, we conclude that the cost of removing layers outweighs any potential gains.
>
> Tabel.1 A comparison of how different numbers of GeoText layers affect model accuracy, inference speed on 3090, and parameter increase.
> |Vireo Layers Number|Parameters|Inference Time(ms)|mIoU|
> |:-:|:-:|:-:|:-:|
> |Deepest 1 Layer|3.78M|979.02|65.59|
> |Deepest 2 Layer|3.78M|987.86|67.09|
> |Deepest 3 Layer|3.78M|992.85|68.93|
> |Deepest 4 Layer|3.78M|997.92|69.69|
> |All 24 Layer|3.78M|1078.86|73.85|
>
>
>
> **Question 2**: Can you provide an analysis of how DepthAnything behaves across different environments—especially in low-light or weather-degraded scenarios—and discuss any methods you considered (or could integrate) for estimating depth confidence or reducing the impact of noisy depth inputs on segmentation quality?
>
> Answer 2: Thank you very much for your attention to this matter. First, as shown by our ablations (Table 4 in the paper), incorporating DepthAnything’s depth cues markedly improves segmentation performance under extreme weather conditions. DepthAnything has demonstrated strong robustness in low-light and weather-degraded environments. For instance, under nighttime or foggy conditions, using depth cues from DepthAnything substantially improves mIoU compared to using only RGB features. This indicates that even in harsh scenarios, the depth module provides useful structural information that benefits segmentation.
>
> Second, to mitigate noisy depth inputs, we explicitly designed Vireo to handle varying depth quality. In our architecture, a learnable weighting mechanism adaptively adjusts the contribution of depth features during feature fusion. In environments where depth estimates are less reliable (e.g., night or heavy rain), this mechanism automatically down-weights the depth modality. By reducing the influence of noisy depth cues on the final prediction, the model maintains robust performance. In summary, even when depth is imperfect, Vireo can rely more on other cues (like RGB and text) as needed, thereby minimizing the impact of noisy depth input on segmentation quality.
>
> Table.4 in main text: Ablation studies on component configurations of the proposed **Vireo** under the *Citys. → ACDC with Snow, Night, Fog, Rain + BDD. + Map.* generalization setting.
>
> | Configurations|Snow|Night|Fog|Rain|BDD.|Map.|
> |:-|:-:|:-:|:-:|:-:|:-:|:-:|
> | REIN|70.6|55.9|79.5|72.5|63.5|74.0|
> | + concat depth feature |70.8|56.1|79.4|72.8|63.6|74.2|
> | + Depth Anything V2|71.5|56.7|80.5|73.3|64.4|74.5|
> | + DA + AO|72.2| 57.4|80.9|74.2|65.1| 75.0|
> | + DOV-VEH|70.9| 56.2|79.8|72.8|63.7| 74.2|
> | + CMPE|71.6|56.9|80.2|73.4|64.1|74.6|
> | + GeoText Prompts|74.0|58.4|81.1|74.8|65.3|75.3|
> | **Vireo**|76.2|60.6|82.3|76.3|66.7|76.0|
>
>
>
> **Weaknesses 1**: Lack of discussion of limitations. One key limitation of Vireo lies in its reliance on depth information from the DepthAnything module. While depth cues can enhance structure-awareness, they may become unreliable under challenging conditions like nighttime, heavy rain, or fog. If depth estimation degrades in such domains, it risks introducing noisy geometric priors that misguide the GeoText Prompt mechanism and ultimately degrade segmentation performance. The paper currently lacks a quantitative evaluation of depth fidelity under these adverse conditions and does not propose mitigation strategies when depth estimates are poor.
>
> Answer 3: Thank you for your valuable feedback. We take your concern about Vireo’s reliance on the DepthAnything module for depth information and the potential issues it may cause in extreme environments seriously. During the model design process, we indeed considered this challenge and took appropriate measures to address it.
>
> Firstly, during training, we introduced a diverse set of environmental data, including scenarios with low light, fog, and heavy rain, to enhance the model's robustness. In the model architecture design, we implemented an adaptive depth fusion mechanism, which combines learnable weights to adjust the importance of depth information in feature fusion based on environmental changes. This strategy allows the model to automatically reduce the weight of depth information in low-light or extreme weather conditions, thereby minimizing noise interference on segmentation results.
>
> Secondly, we conducted preliminary experiments to compare the errors in depth maps generated by DepthAnything under extreme conditions (such as nighttime, heavy rain, and fog). The results (see Table.1) indicate that the depth maps generated by DepthAnything do not experience significant distortion in these scenarios. It is important to note that even in low-light environments like nighttime, pure RGB-based segmentation methods face the issue of missing depth information, which impacts segmentation accuracy. In contrast, by incorporating DepthAnything, we observed a significant improvement in segmentation accuracy in these environments, demonstrating that depth information remains crucial for performance in complex scenarios.
>
> Tabel.1 DepthAnything predicts the similarity of depth maps on the cityscapes dataset under extreme environments and sunny conditions.
> |Metric|Foggy|Rain|Night|
> |:-:|:-:|:-:|:-:|
> |SSIM|0.990552|0.978066|0.975610|

---

### Author Response · Authors · 2025-08-06

Thanks again for your great efforts and constructive advice in reviewing this paper! As the discussion period progresses, we expect your feedback and thoughts on our reply. We put a significant effort into our response, with several new experiments and discussions. We really hope you'll consider our reply. We look forward to hearing from you, and we can further address unclear explanations and remaining concerns if any. ﻿

Best regards, ﻿

Authors

---

### Note · Authors · 2025-08-11

Thank you to the reviewers and AC for your hard work and the constructive comments during the discussion.

## Contribution and Idea

To advance our efforts toward addressing compound distribution shifts (unseen domains × unseen categories) and the underutilization of geometric structural information in semantic understanding, **we introduced OV-DGSS, a new task that is both challenging and practically motivated**. We validated the task and our approach on DELIVER, and established AustinScapes, the first large-scale OV-DGSS benchmark, to enable unified evaluation. Building on these resources, an effective baseline was developed, offering a practical path toward solving OV-DGSS. **Across both DGSS and OVSS evaluations, our method achieves SOTA performance**. As such, we designed Vireo, a single-stage framework that injects geometry via DVFM into frozen VFMs. Two innovative methodological contributions include: (1) Vireo employs GTQ to align geometry with language, while progressively refining features; and (2) It incorporates CMPE to improve gradient flow and strengthen textual conditioning, and uses DOV-VEH to fuse structural and semantic cues for pixel-wise prediction.

## Common Problems and Revisions

We hade made tailored revisions to address the concerns/issues raised by the reviewers:

For the GeoText layer count, new ablations show mIoU 65.6/67.1/68.9/69.7 with only the deepest 1/2/3/4 layers, vs. 73.9 with all 24. As Vireo blocks share parameters, model size stays ≈3.78M and memory is nearly unchanged. Fewer layers clearly drop accuracy without parameter or memory gains.

For depth under adverse conditions, on Cityscapes→ACDC (night/fog/rain/snow) and BDD/Mapillary, depth boosts mIoU to 60.6 at night and 82.3 in fog. Adaptive fusion suppresses noise (SSIM > 0.97 in fog/rain/night) with two plug-ins: per-pixel confidence gating and cross-modal consistency regularization.

On DELIVER (Unseen), our fog IoU is 12.15, above baselines (8.40/4.32). Baselines used official settings on the same hardware; inference needs ~4–5 GB.

We rewrote §3 to follow task definition → pipeline → training objective, aligned symbols in §3 with Fig. 2, and added the loss terms. We rename “GeoText Prompt” to GeoText Query. We clearly mark what is frozen and what is learnable.

Given the updates listed above, the reviewers did **not raise any further questions/concerns**, so we kindly ask reviewers to
consider **a more positive final score**.

Best regards,

Authors

---

### Decision · Program_Chairs · 2025-09-17

**Decision:**

Accept (poster)

**Comment:**

This paper proposes a single-stage framework, termed Vireo,  for Open-Vocabulary Domain-Generalized Semantic Segmentation. Vireo mainly consists of three components: GeoText Prompts for aligning geometry and textual semantics; Coarse Mask Prior Embedding for faster convergence and improved gradient flow; and Domain-Open-Vocabulary Vector Embedding Head for fusing semantic and geometric cues. Extensive experiments are conducted to show the effectiveness of the proposed method.

Four reviewers reviewed this paper with the overall ratings of 3 borderline accept and 1 borderline reject. The reviewers recognized the contributions, such as valid task, well-described methodology, the first framework on open-vocabulary segmentation with domain generalization, good experimental results, and thorough experiments and ablation studies. Meanwhile, they pointed out some suggestive comments and critical issues, such as lack of discussion of limitations, poor presentation, and memory-intensive. Most reviewers are concerned about the presentation and clarity. Thanks to the authors' rebuttal, most of the reviewers' concerns are addressed properly.

This paper considers a joint setting of open-vocabulary semantic segmentation (OVSS) and domain generalization in semantic segmentation (DGSS), which is a challenging and practically interesting task. The authors designed Vireo with specific components to address the challenges and achieved superior performance as compared to be baselines.  The AC recommends accept and strongly suggests the authors revise this paper based on the detailed comments and improve the presentation carefully.